# Genome-wide meta-analysis identifies novel risk loci for uterine fibroids within and across multiple ancestry groups

Jeewoo Kim [1,2,3,4], Ariel Williams[5], Hannah Noh [6,7], Elizabeth A. Jasper[1,8], Sarah H. Jones[9], James A. Jaworski[4,10], Megan M. Shuey [3], Edward A. Ruiz-Narváez [11], Lauren A. Wise[12], Julie R. Palmer[13], John Connolly[14], Jacob M. Keaton[5,10], Joshua C. Denny [5,15], Atlas Khan [16], Mohammad A. Abbass[17], Laura J. Rasmussen-Torvik [18], Leah C. Kottyan [19], Purnima Madhivanan[20,21,22,23], Karl Krupp[20,22,24], Wei-Qi Wei [8], Todd L. Edwards [10,26], Digna R. Velez Edwards[1,8,9,25,26,27] ✉ & Jacklyn N. Hellwege [3,4,25,26,27] ✉

Uterine leiomyomata or fibroids are highly heritable, common, and benign tumors of the uterus with poorly understood etiology. Previous GWAS have reported 72 associated genes but included limited numbers of non-European individuals. Here, we identify 11 novel genes associated with fibroids across multi-ancestry and ancestry-stratified GWAS analyses. We replicate a known fibroid GWAS gene in African ancestry individuals and estimate the SNP-based heritability of fibroids in African ancestry populations as 15.9%. Using genetically predicted gene expression and colocalization analyses, we identify 46 novel genes associated with fibroids. These genes are significantly enriched in cancer, cell death and survival, reproductive system disease, and cellular growth and proliferation networks. We also find that increased predicted expression of *HEATR3* in uterine tissue is associated with fibroids across ancestry strata. Overall, we report genetic variants associated with fibroids coupled with functional and gene pathway enrichment analyses.

Uterine fibroids, or leiomyomata, are benign monoclonal elliptical growths of smooth muscle and connective tissue in the myometrium of the uterus. They are the most common benign tumor affecting people with a uterus. The cumulative incidence of fibroids by age 50 is estimated to be almost 70% among White individuals and over 80% in Black individuals[1]. Common symptoms of fibroids include heavy menstrual bleeding, abnormal menstrual bleeding, and pelvic pain[2]. People with fibroids report quality-of-life scores similar to or lower than patients with other chronic diseases such as, diabetes, breast cancer, and cardiovascular disease[3]. Fibroids are the leading cause of hysterectomy, comprising 40% of indications[4], and the estimated annual costs for the United States due to fibroids are between $5.9 to $34.4 billion[5]. Established risk factors for fibroids include early

menarche, obesity, premenopausal age, Black race (self-selected), and family history[6]. A protective effect has been seen with parity[1,7–9]. Additional factors that have been associated with fibroids include, and are not limited to, vitamin D deficiency[10,11], hypertension[12,13], stressful life events[14–16], and experiences of racial discrimination[17].

Genetic factors also contribute to risk, with estimated twin-based heritability ranging from 26 to 63%[18,19]. There have been 13 fibroid genome-wide association studies (GWAS) that collectively report 72 genes with genome-wide significant variants[20–32]. Investigations of fibroid genetics in diverse ancestry groups have been limited in African ancestry populations[20,24].

Here we report a large, multi-ancestry GWAS meta-analysis of uterine fibroids including 74,294 cases (27.7% non-European descent)

and 465,810 controls (18.3% non-European descent) from a combination of publicly available summary statistics and newly available data. We identified multiple novel genes associated with fibroids in both multi-ancestry and ancestry-specific analyses. We report functional interpretation of significant loci with genetically predicted gene expression. Genes were also characterized into functional pathways and enrichment sets, and pathways relating to cancer and cell cycle were observed.

## Results

### Data summary

The data used in these meta-analyses come from a combination of publicly-available summary statistics and newly run GWAS (Fig. 1; Supplementary Data 1)[20,24,27,33]. We conducted four meta-analyses: 1) The European ancestry meta-analysis comprised of 53,711 cases and 380,441 controls. 2) The East Asian/Central South Asian ancestry meta-analysis data comprised of 14,905 cases and 69,609 controls. 3) The African ancestry meta-analysis data comprised of 5678 cases and 15,760 controls. 4) The multi-ancestry meta-analysis comprised of 74,294 cases and 465,810 controls. Variants described below were mapped by nearest distance to transcription start sites.

### Multi-ancestry and ancestry stratified GWAS meta-analyses SNP-level associations

The results of the multi-ancestry meta-analysis are summarized in Fig. 2 and Supplementary Data 2a, 3a. There were 372 sentinel single nucleotide polymorphisms (SNPs) associated with fibroids in the multi-ancestry meta-analysis (independent significant SNPs with $r^2 > 0.1$) (Supplementary Data 2a). The overall most significant SNPs were rs78378222 (odds ratio [OR] 0.53, 95% confidence interval [CI] 0.50–0.56, $p = 2.57 \times 10^{-132}$) which maps to gene *TP53* (3 prime untranslated region [UTR]), and rs58415480 (OR 0.82, 95% CI 0.81–0.84, $p = 5.58 \times 10^{-115}$) which maps to *SYNE1* (intron), both well-established associations with fibroids[21,24,27,28]. After conditional analyses, we identified eight fibroid-associated genes with at least one significantly associated variant that have not been previously published (Fig. 2, Supplementary Data 3a, Supplementary Fig. 6a–f). In our discussion of results, novel associations refer to genes that have never been reported to be associated with fibroids, while "previously unpublished" gene associations include those that have been reported in biobank databases but have not been described in literature. The two novel gene associations include rs74582999 (*VIP*, intron) and rs761779 (*FOXO3*, regulatory region). The other six previously unpublished genes include rs149261442 (*TEKT1*, intron), rs184210518 (*SLC16A11*, downstream), rs184918809 (*RPEL1*, upstream), rs555566736 (*SLC12A7*, intron), rs76382168 (*TTC28*, intron) and rs78899396 (*POLR2A*, intron). Of note, rs76382168 has a heterogeneity *p*-value less than 0.01. We identified an

additional 22 significant secondary signals after conditioning on the 372 sentinel SNPs (Supplementary Data 2a). We validated a total of 190 variants in the GWAS Catalog from previous fibroid GWAS.

The genomic inflation factor $\lambda_{GC}$ was 1.09 (Supplementary Fig. 1a), and the Linkage Disequilibrium Score Regression (LDSC) intercept was 1.06 (standard error [SE] 0.01; Supplementary Data 4). The SumHer Linkage Disequilibrium Adjusted Kinships (LDAK) model SNP-based heritability method was used to estimate heritability for each meta-analysis[34]. The multi-ancestry meta-analyses heritability was estimated using the GBR HapMap3 annotated tags, and heritability was estimated at 0.05 (SE 0.002; Supplementary Data 4).

The results of the European ancestry meta-analysis are summarized in Fig. 3 and Supplementary Data 2b, 3b. In the European ancestry analysis, there were 216 sentinel SNPs associated with fibroids (Supplementary Data 2b). The overall most significant SNPs were the same as the multi-ancestry, rs78378222 (OR 0.53, 95% CI 0.50–0.56, $p = 2.57 \times 10^{-132}$, *TP53*, 3 prime UTR), and rs58415480 (OR 0.82, 95% CI 0.80–0.83, $p = 2.67 \times 10^{-113}$, *SYNE1*, intron). The effect size and *p*-value for rs78378222 are the same as the multi-ancestry because only the European data cohorts contributed to this variant in the multi-ancestry analysis. After conditional analyses, we identified four fibroid-associated genes with at least one significantly associated variant that have not been described in prior studies (Fig. 3, Supplementary Data 3b, Supplementary Fig. 6g–j). The two novel gene associations included rs74582999 (*VIP*, intron) and rs76798800 (*DCST2*, intron). The two previously unpublished genes included rs149261442 (*TEKT1*, intron) and rs35447002 (*C11orf87*, intergenic). Of note, rs149261442 and rs35447002 have a heterogeneity *p*-value less than 0.01. We identified an additional 11 significant secondary signals after conditioning on the 216 sentinel SNPs (Supplementary Data 2b). We validated a total of 178 variants in the GWAS Catalog from previous fibroid GWAS. For the European analysis the $\lambda_{GC}$ was 1.17 (Supplementary Fig. 1b), and the LDSC intercept was 1.07 (SE 0.011; Supplementary Data 4)[35]. Using GBR HapMap3 annotated tags, heritability was estimated to be 0.07 (SE 0.003).

The results of the East Asian/Central South Asian ancestry meta-analysis are summarized in Fig. 4 and Supplementary Data 2c, 3c. In the East Asian/Central South Asian ancestry analysis there were 108 sentinel SNPs (Supplementary Data 2c). The overall most significant SNPs were rs73392700 (OR 0.77, 95% CI 0.74–0.79, $p = 1.09 \times 10^{-50}$) in *SIRT3* (intron) and rs141244868 (OR 0.77, 95% CI 0.74–0.79, $p = 1.30 \times 10^{-50}$) in *PSMD13* (intron), and both genes have been previously associated with fibroids[23,24,27,30]. We identified an additional two significant variants after conditioning on the 108 sentinel SNPs (Supplementary Data 2c). No novel gene associations were identified, but we did identify secondary signals in previously associated genes (Supplementary Data 3c). We validated a total of 110 variants in the GWAS

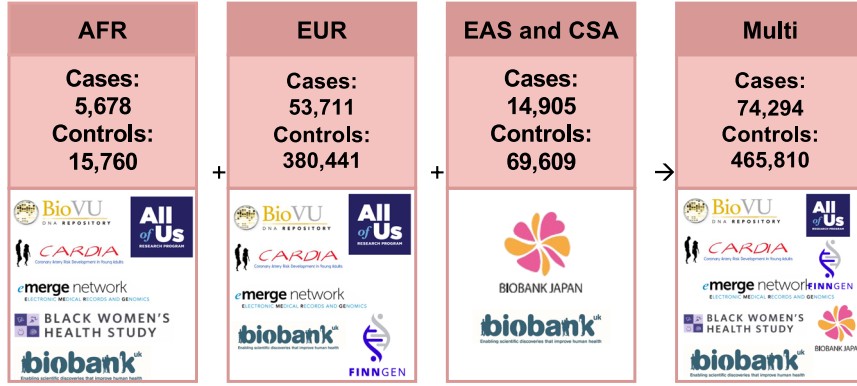

**Fig. 1 | Cohort sources and case/control counts for each meta-analysis.** AFR African Ancestry, EUR European Ancestry, EAS East Asian Ancestry, CSA Central South Asian Ancestry, Multi Multi Ancestry.

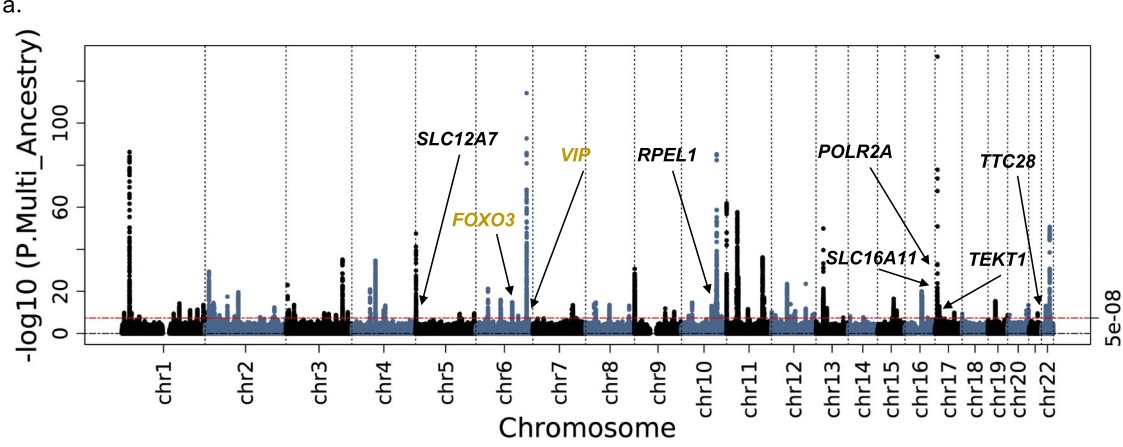

**Fig. 2 | Multi-ancestry genome-wide meta-analysis identified eight genes associated with fibroids unpublished in previous literature. a** Manhattan plot, gold labels = novel, black labels = previously unpublished. **b** Sentinel variant at novel gene results, asterisk = novel gene association. Logistic regression statistical tests; multiple testing correction *p*-value threshold used ($5 \times 10^{-8}$).

Catalog. Within East Asian/Central South Asian ancestry analysis the $\lambda_{GC}$ was 1.07 (Supplementary Fig. 1c), and the LDSC intercept was 1.01 (SE 0.009; Supplementary Data 4). Using SumHer and the EAS HapMap3 tags, heritability was estimated to be 0.115 (SE 0.007).

The results of the African ancestry meta-analysis are summarized in Fig. 5 and Supplementary Data 2d, 3d. In the African ancestry analysis, there were two sentinel SNPs and a total of six statistically significant SNPs. The most significant variant was rs56897532 (OR 0.78, 95% CI 0.72–0.85, *p* = $5.39 \times 10^{-9}$) in *COL22A1* (intergenic) and is a novel gene association with fibroids (Fig. 5, Supplementary Data 3d, Supplementary Fig. 6k). Variants rs12280757, rs7106353, rs12285041, rs12285028, and rs61889170 map to *WT1* (all intronic) and were the next most significant. *WT1* has been previously associated with fibroids but not in this population. These specific variants are different from the previously identified variants in *WT1*[21,24,27,28]. The association with the variant in *COL22A1* was only observed in the African ancestry analysis. The $\lambda_{GC}$ was 1.01 (Supplementary Fig. 1d), and the LDSC intercept was 1.01 (SE 0.006, Supplementary Data 4). Using AFR HapMap3 tags, heritability was estimated to be 0.159 (SE 0.040).

A detailed overview of effect sizes and effect allele frequency plots between the ancestry groups are shown in Supplementary Fig. 2. The effect size correlations of all significant independent variants from any of the four meta-analyses are plotted. Between the African ancestry and the European ancestry analyses, the effect size correlation (R) was 0.44. The correlation between the East Asian/Central South Asian ancestry and the European ancestry meta-analyses was 0.92. The effect allele frequency correlation for both the African and European ancestries analyses (R = 0.62) and the East Asian/Central South Asian and European ancestries analyses (R = 0.95) were positive.

### Functional annotation and gene set analysis
We used the Functional Mapping and Annotation (FUMA) web tool[36] to conduct various post-GWAS analyses (Supplementary Data 5, 6;

Supplementary Fig. 3). In the multi-ancestry, European ancestry, and East Asian/Central South Asian ancestry analyses, we found that associated variants were primarily intronic with significant enrichment in this variant category, and also statistically significant depletion in intergenic and non-coding RNA regions, consistent with previous research (Supplementary Fig. 3a–c)[31]. The associated variants within African ancestry analysis were significantly enriched in intergenic regions and non-coding RNA intronic regions (Supplementary Fig. 3d).

A gene-set enrichment test was also performed by FUMA for 476 input genes. For the multi-ancestry analysis, the top five significantly enriched Gene Ontology biological processes gene sets were replicative senescence (adjusted *p* = $4.74 \times 10^{-4}$), DNA damage response signal transduction by p53 class mediator resulting in cell cycle arrest (adjusted *p* = $4.74 \times 10^{-4}$), mitotic G1 S transition checkpoint signaling (adjusted *p* = $6.08 \times 10^{-4}$), regulation of response to DNA damage stimulus (adjusted *p* = $6.38 \times 10^{-4}$), and negative regulation of telomere maintenance (adjusted *p* = $1.87 \times 10^{-3}$). There were 52 other significantly enriched Gene Ontology biological processes gene sets, with many related to cell cycle control, DNA damage response, and p53 signaling (Supplementary Data 5a). There were 110 GWAS catalog phenotypes with significant gene set enrichment as well. Uterine fibroids were the most significant (adjusted *p* = $2.10 \times 10^{-55}$). The next four most significantly enriched gene-sets were refractive error (adjusted *p* = $6.23 \times 10^{-17}$), waist-to-hip ratio adjusted for BMI (adjusted *p* = $6.46 \times 10^{-14}$), prostate cancer (adjusted *p* = $1.41 \times 10^{-13}$), and Crohn's disease (adjusted *p* = $2.07 \times 10^{-10}$) (Supplementary Data 6a).

The enrichment tests with 378 input genes in the European ancestry group had 10 significant Gene Ontology biological processes gene sets, with the top five being replicative senescence (adjusted *p* = $4.87 \times 10^{-4}$), post-transcriptional regulation of gene expression (adjusted *p* = $1.49 \times 10^{-2}$), cell cycle G1 S phase transition (adjusted *p* = $1.49 \times 10^{-2}$), signal transduction involved in cell cycle checkpoint (adjusted *p* = $1.58 \times 10^{-2}$), and mitotic cell cycle (adjusted *p* = $1.58 \times 10^{-2}$)

a.

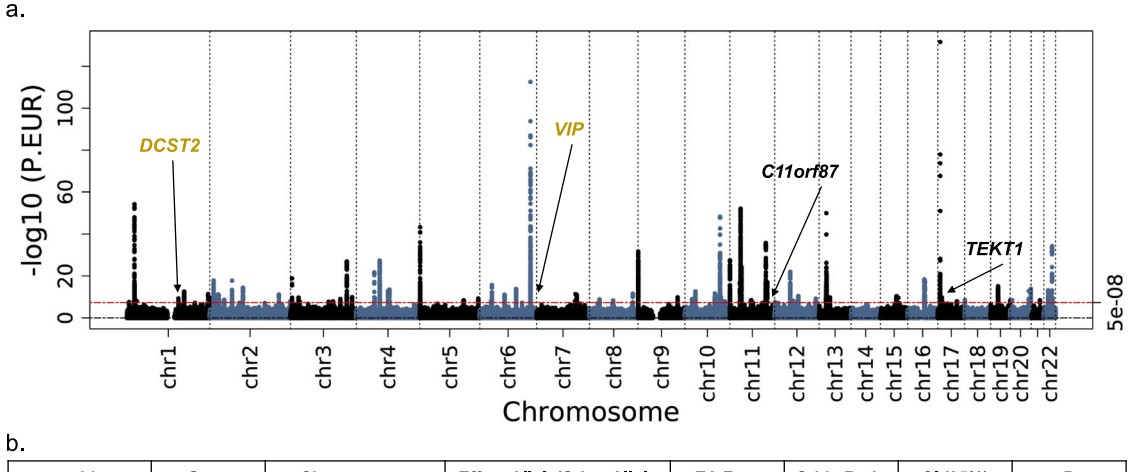

b.

| rsid | Gene | Chromosome | Effect Allele/Other Allele | EA Freq | Odds Ratio | CI (95%) | P |
|---|---|---|---|---|---|---|---|
| rs76798800 | DCST2* | 1 | T/G | 0.247 | 1.05 | 1.04-1.07 | $5.59 \times 10^{-10}$ |
| rs74582999 | VIP* | 6 | T/C | 0.966 | 0.88 | 0.85-0.92 | $2.07 \times 10^{-08}$ |
| rs35447002 | C11orf87 | 11 | T/C | 0.222 | 0.94 | 0.93-0.96 | $2.28 \times 10^{-11}$ |
| rs149261442 | TEKT1 | 17 | C/G | 0.992 | 0.75 | 0.69-0.82 | $2.12 \times 10^{-11}$ |

**Fig. 3 | European ancestry genome-wide meta-analysis identified four genes associated with fibroids unpublished in previous literature. a** Manhattan plot, gold labels = novel gene associations, black labels = previously unpublished genes. **b** Sentinel variant at novel gene results, asterisk = novel gene association. Logistic regression statistical tests; multiple testing correction $p$-value threshold used ($5 \times 10^{-8}$).

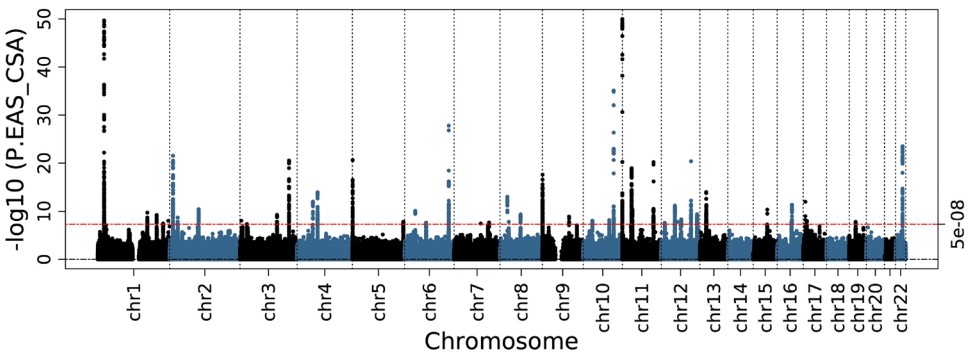

**Fig. 4 | East Asian/Central South Asian ancestry genome-wide meta-analysis results.** Significant genes were replications of previously identified associations. Logistic regression statistical tests; multiple testing correction $p$-value threshold used ($5 \times 10^{-8}$).

(Supplementary Data 5b). There were 81 GWAS catalog reported gene-sets that had statistically significant enrichment. Again, uterine fibroids were the most significant (adjusted $p = 2.00 \times 10^{-44}$) followed by prostate cancer (adjusted $p = 2.16 \times 10^{-22}$), Crohn's disease (adjusted $p = 1.22 \times 10^{-10}$), and pancreatic cancer (adjusted $p = 8.72 \times 10^{-9}$) (Supplementary Data 6b).

In the East Asian/Central South Asian ancestry analysis, with 124 input genes, there were 12 significant Gene Ontology biological processes enrichment sets. Similar to the previous two, they were DNA damage response signal transduction by p53 class mediator resulting in cell cycle arrest (adjusted $p = 6.50 \times 10^{-3}$), hepoxilin metabolic process (adjusted $p = 2.19 \times 10^{-2}$), replicative senescence (adjusted $p = 2.19 \times 10^{-2}$), DNA damage response signal transduction by p53 class mediator (adjusted $p = 2.19 \times 10^{-2}$), and mitotic G1 S transition checkpoint signaling (adjusted $p = 2.31 \times 10^{-2}$) (Supplementary Data 5c). There were 47 significant GWAS catalog gene sets enriched, where the top five were uterine fibroids (adjusted $p = 3.02 \times 10^{-48}$), waist-to-hip ratio adjusted for BMI (adjusted $p = 4.39 \times 10^{-8}$), waist-hip index (adjusted $p = 2.01 \times 10^{-7}$), platelet

count (adjusted $p = 2.10 \times 10^{-6}$), and gynecologic disease (adjusted $p = 2.10 \times 10^{-6}$) (Supplementary Data 6c).

In the multi-ancestry analysis, there was significant enrichment in upregulated differentially expressed gene sets for uterine tissue (Supplementary Fig. 3a). For the European ancestry analyses, there was significant enrichment in upregulated differentially expressed gene sets for uterine tissue and cervical uterine tissue (Supplementary Fig. 3b). East Asian/Central South Asian ancestry analyses had no significant enrichment (Supplementary Fig. 3c). We were underpowered to apply these gene-based tests to the African ancestry meta-analysis results.

**Transcriptome-wide association analysis with genetically predicted gene expression**

We used S-PrediXcan with 49 tissues from the GTEx project to interpret the associations between genetically predicted gene expression (GPGE) and uterine fibroid risk (Supplementary Fig. 4a). In the multi-ancestry analysis, we identified 1223 significant gene expression-tissue pairs ($p$-value $< 9.4 \times 10^{-8}$) associated with fibroids across all tissues, of

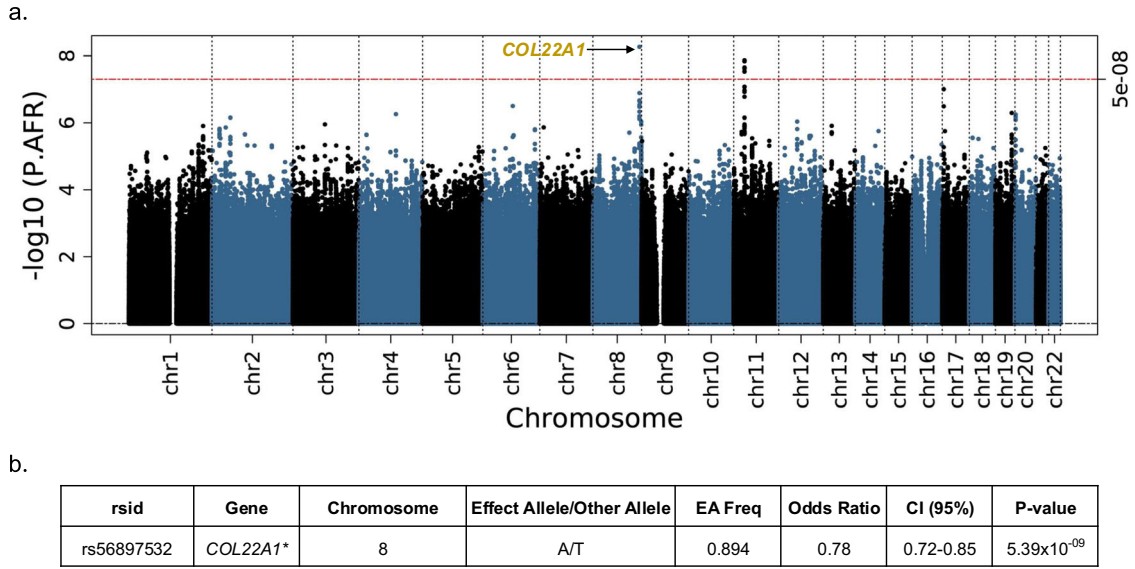

**Fig. 5 | African ancestry genome-wide meta-analysis identified one novel gene associated with fibroids. a** Manhattan plot, gold labels = novel gene associations.
**b** Sentinel variant at novel gene result, asterisk = novel gene association. Logistic regression statistical tests; multiple testing correction *p*-value threshold used ($5 \times 10^{-8}$).

which 240 had a colocalized SNP (H4 posterior probability > 0.8, H3 posterior probability < 0.3) (Supplementary Data 7a). Of these, there were 60 unique genes and 40 of those genes have not been previously associated with fibroids in the GWAS catalog. The most significant colocalized GPGE association was with *ACTRT3* expression in the pituitary tissue (OR 5.54, 95% CI 4.19–7.33, $p = 3.43 \times 10^{-33}$). *HEATR3* predicted gene expression had a significant and colocalized association where increased predicted expression consistently associated with increased fibroid risk.

In ancestry-specific analyses, we identified multiple significant predicted gene expression associations (Supplementary Data 7b–d, Supplementary Fig. 4b–d), some of which were unique significant gene-tissue colocalized pairs not observed in the multi-ancestry analysis (Table 1). In the European analysis, we identified 31 significant novel and colocalized unique gene associations with fibroids (Supplementary Data 7b), four of which were not significant and colocalized in other analysis strata. There were 16 significant novel colocalized unique gene associations with fibroids in the East Asian/Central South Asian ancestry analysis (Supplementary Data 7b). The most significant association was *WNT4* in lung tissue (OR 0.43, 95% CI 0.37–0.49, $p = 6.01 \times 10^{-32}$).

In the African ancestry analysis, we identified 15 significant gene expression-tissue pairs associated with fibroids across all tissues, but none had a colocalized gene (Supplementary Fig. 4d). Within these 15 pairs, there were four unique genes (*NQO1, AC004889.1, TXNL4B,* and *GGT1*), and all four of those genes have not been previously associated with fibroids in the GWAS catalog and were significant only in the African ancestry analysis (Supplementary Data 7d). The most significant GPGE association was with *NQO1* in esophagus mucosa (OR 0.67, 95% CI 0.60–0.76, $p = 3.39 \times 10^{-10}$). In sum, we identified 46 novel significant and colocalized genes that have not previously been associated with fibroids across all four analyses.

Within specifically uterine tissue for the multi-ancestry analysis, there were 13 significant GPGE associations, and the only colocalized one was at *HEATR3* (Table 1, Supplementary Fig. 5a). *HEATR3* has been previously associated in a GPGE analysis, but in skeletal tissue only[24]. Seven of the 13 are novel gene associations with fibroids. Within uterine tissue for the European ancestry analysis, there were 15 significant GPGE associations, and the one that colocalized was also at *HEATR3* (Table 1, Supplementary Fig. 5b). 11 of these 15 associations are novel gene associations with fibroids. The 11 novel significant gene

associations observed in the European analysis included *SULT1E1, LRRC34, ATP1B2, LYG1, MIR1915HG, NPAT, UIMC1, LINC01126, AC010883.1, UTP3,* and *C12orf65*. For the East Asian/Central South Asian ancestry uterine tissue analysis, there were six significant GPGE associations (Supplementary Fig. 5c). Two of these are novel gene associations with fibroids, *CDK2AP1* and *C12orf65*, and again *HEATR3* was the only gene with a colocalized result (Table 1). *CDK2AP1* predicted expression in uterine tissue was only significantly associated in the East Asian/Central South Asian analysis. There were no significant predicted gene expression associations in uterine tissue in the African ancestry analysis (Supplementary Fig. 5d).

### Ingenuity Pathway Analysis
We ran Ingenuity Pathway Analysis using all colocalized significant GPGE results across the four meta-analyses and incorporating multiple open access gene map and pathway databases. The top three networks that had significant enrichment of overlapping molecules included (1) Cancer, Organismal Injury and Abnormalities, Reproductive System Disease; (2) Cancer, Neurological Disease, Organismal Injury and Abnormalities; and (3) Cell Death and Survival, Cellular Movement, Organismal Injury and Abnormalities (Supplementary Data 8). The molecular connectivity of the first network is illustrated in Fig. 6, with primary molecules including 16 focus molecules such as *BET1L* forming the foundation of the network. Out of 400 pathways tested, 75 were significant. The top enriched pathways included Ovarian Cancer Signaling ($-\log(p) = 3.05$), HOTAIR Regulatory Pathways ($-\log(p) = 3.45$), and SUMOylation of transcription factors ($-\log(p) = 3.13$) (Supplementary Data 9). The two most significant upstream regulators, calculated from experimentally observed relationships between regulators and dataset genes to predict upstream transcriptional regulators, were *TRRAP* and *TFAP2A*. (Supplementary Data 10).

## Discussion
We present results from multi-ancestry genome-wide meta-analyses of uterine fibroids that leverages a larger sample size relative to prior studies, inclusion of diverse populations, and assessment of GPGE and pathways analysis of results. Within this study, we identified a total of 11 previously unpublished fibroid-associated genes. Our SNP-based heritability estimates from all four analyses were in the range of previously reported heritability (2.8–33%)[28,31,37], and we observed high heritability in the African ancestry analysis (15.9%). We used

**Table 1 | Significant predicted gene expression analyses specific to uterine tissue and ancestry analyses**

| Gene Name | Odds Ratio | 95% Confidence Interval | P-Value | Tissue |
|---|---|---|---|---|
| **Shared Signal (Uterine Tissue)** | | | | |
| HEATR3 | 1.07 | 1.05–1.08 | $3.40 \times 10^{-19}$ | Uterus Multi Ancestry |
| HEATR3 | 1.08 | 1.06–1.10 | $6.46 \times 10^{-18}$ | Uterus EUR |
| HEATR3 | 1.06 | 1.04–1.09 | $2.91 \times 10^{-08}$ | Uterus EAS |
| **European Analysis-Specific** | | | | |
| AC079906.1* | 1.16 | 1.12–1.21 | $3.01 \times 10^{-14}$ | Ovary |
| SULT1B1 | 0.85 | 0.81–0.89 | $1.81 \times 10^{-11}$ | Testis |
| NEK10 | 0.58 | 0.49–0.69 | $6.49 \times 10^{-10}$ | Esophagus Muscularis |
| PIAS1* | 0.85 | 0.81–0.90 | $1.01 \times 10^{-09}$ | Muscle Skeletal |
| MIR1915HG* | 1.16 | 1.11–1.23 | $9.97 \times 10^{-09}$ | Esophagus Muscularis |
| AC114730.3* | 0.89 | 0.86–0.93 | $1.48 \times 10^{-08}$ | Pancreas |
| ING5 | 0.92 | 0.89–0.95 | $7.82 \times 10^{-08}$ | Pancreas |
| **East Asian/Central South Asian Analysis-Specific** | | | | |
| WNT4 | 0.43 | 0.37–0.49 | $6.01 \times 10^{-32}$ | Lung |
| TERT | 0.69 | 0.63–0.76 | $8.94 \times 10^{-17}$ | Skin Not Sun Exposed Suprapubic |
| TXNDC9* | 0.92 | 0.89–0.94 | $5.33 \times 10^{-10}$ | Testis |
| PGBD2 | 0.75 | 0.67–0.83 | $8.22 \times 10^{-08}$ | Brain Cerebellum |

(Upper panel) Significant and colocalized gene expression associations with fibroids in uterus tissue shared across multiple analysis groups, (Middle panel) Significant and colocalized gene expression associations unique to the European analysis, and (Lower panel) Significant and colocalized gene expression associations unique to the East Asian/Central South Asian analysis. Most significant gene-tissue pair shown. Asterisk = novel associations with fibroids. Two-sided Wald test; multiple testing correction p-value threshold used ($9.4 \times 10^{-8}$).

an LDAK approach for heritability estimation (SumHer) due to availability of reference annotations tags for each analysis and improved performance in estimates of complex traits[34,38]. We also identified 46 novel genes associated with fibroids from GPGE and colocalization analyses.

We observed correlated effect allele frequencies between both the East Asian/Central South Asian and European ancestry analyses, and the African and European ancestry analyses. The effect directions of most variants were generally consistent between European and East Asian/Central South Asian ancestry populations. There was less consistency with the African ancestry group as evidenced by the correlation R of 0.44 for the European ancestry and African ancestry effect size comparison, which could be attributable to the smaller sample size of the African ancestry analyses (Supplementary Fig. 1a, b).

In our multi-ancestry meta-analysis, we identified six genes associated with fibroids only in publicly available biobank browsers and two novel genes, VIP (vasoactive intestinal peptide), and FOXO3 (forkhead box O3). VIP has been shown to inhibit uterine smooth muscle activity and act as a neurotransmitter in the female genital tract[39]. Additionally, VIP infusion has been shown to reverse fibrosis in myocardial cells in rats[40] and VIP is known to have effects on blood pressure, which as mentioned previously is a fibroid risk factor[41]. FOXO3 is predicted to be involved in neoplastic cell transformation with tumor progression and angiogenesis, and there is conflicting evidence on whether it improves or harms prognosis. FOXO3 has a conserved domain that mediates its interactions between estrogen related receptor α (ERRα) and tumor protein p53[42]. Recently published data suggests that FOXO3a expression is increased in uterine smooth muscle tumors compared to normal myometrium[43].

The DCST2 association was significant only in the European ancestry analysis. The index SNP (rs76798800) has been associated with 25-hydroxyvitamin D concentrations with the effect allele associated with decreased vitamin D levels and increased risk of fibroids[44]. Vitamin D deficiency is a risk factor for fibroids with evidence of overlapping genetic risk loci, and supplementation, and higher concentrations of vitamin D has been shown to decrease the size of fibroids[11,45,46].

A majority of the previously unpublished genes have been associated with tumorigeneses and tumor-suppressor pathways, primarily in cancer phenotypes, including: SLC16A11, POLR2A, C11orf87, and SLC12A7 [47–52]. Some of these genes are associated with other risk factors for fibroids like hypertension, such as SLC12A7[12,53]. TTC28 has been shown to be significantly down-regulated in leiomyomas with certain chromosomal deletions, but not in normal myometrium tissue[54].

African ancestry analyses resulted in two significant independent loci, with variants uniquely significant in this analysis. One of the genes, WT1 (Wilms Tumor 1) is the first replication of a known fibroid-associated gene in African ancestry individuals[24,28]. The other gene, (Collagen Type XXII Alpha 1 Chain, COL22A1), is a novel fibroid association. Differentially expressed gene analysis has identified COL22A1 upregulation in fibroid tumors of all sizes[55]. COL22A1 is a gene that encodes collagen that structurally belongs to the FACIT protein family (fibril-associated collagens with interrupted triple helices).

We identified many significant associations of gene expression in specific tissues with fibroids through GPGE analyses. One notable association with a high odds ratio was ACTRT3 (actin related protein T3) in pituitary tissue (OR 5.54, 95% CI 4.19–7.33, $p = 3.43 \times 10^{-33}$). ACTRT3 expression has been observed to be increased in bladder tumors[56]. We identified one significant and colocalized gene in uterine tissue, HEATR3 (HEAT repeat protein 3), with an increased gene expression associated with risk of fibroids. This gene has been identified in previous S-PrediXcan studies but was only significantly associated with predicted expression in skeletal tissue[24]. HEATR3 has an important role in ribosomal protein transport and 5S ribonucleoprotein particle assembly[57]. Other studies have identified this as a risk gene associated with esophageal cancer and have observed an increased expression of HEATR3 in bladder cancer[58,59]. Further research into the role of increased HEATR3 gene expression in fibroid pathogenesis may illuminate future therapeutic directions.

In the multi-ancestry and European ancestry FUMA pathway enrichment analyses, we observed significant associations for Gene Ontology biological pathways of DNA damage response, cell cycle signaling, and checkpoint signaling. Our East Asian/Central South Asian enriched pathways included multiple pathways relating to gonad development and sex determination, and the others relating to smooth muscle cell migration and proliferation. However, all three analysis groups had the most significant GWAS catalog enrichment in uterine fibroids. These analyses provide a way to interpret the differentially powered tests. The collective and combination of results may be more informative rather than comparing the results, as differentially enriched pathways across ancestry groups is likely a function of power, and less of differing biology.

We used use Ingenuity Pathway Analysis to characterize our many significant predicted gene expression results across all the analysis groups. Shared central molecules of some of the most significant pathways included CD44, WNT4, or TP53. The significant networks, similar to the patterns observed in the FUMA pathway enrichment analyses, centered around cancer and cell cycle, with the most significant network being related to reproductive system disease.

Despite our study representing a large increase in both sample size and diversity of data used to analyze associations with uterine

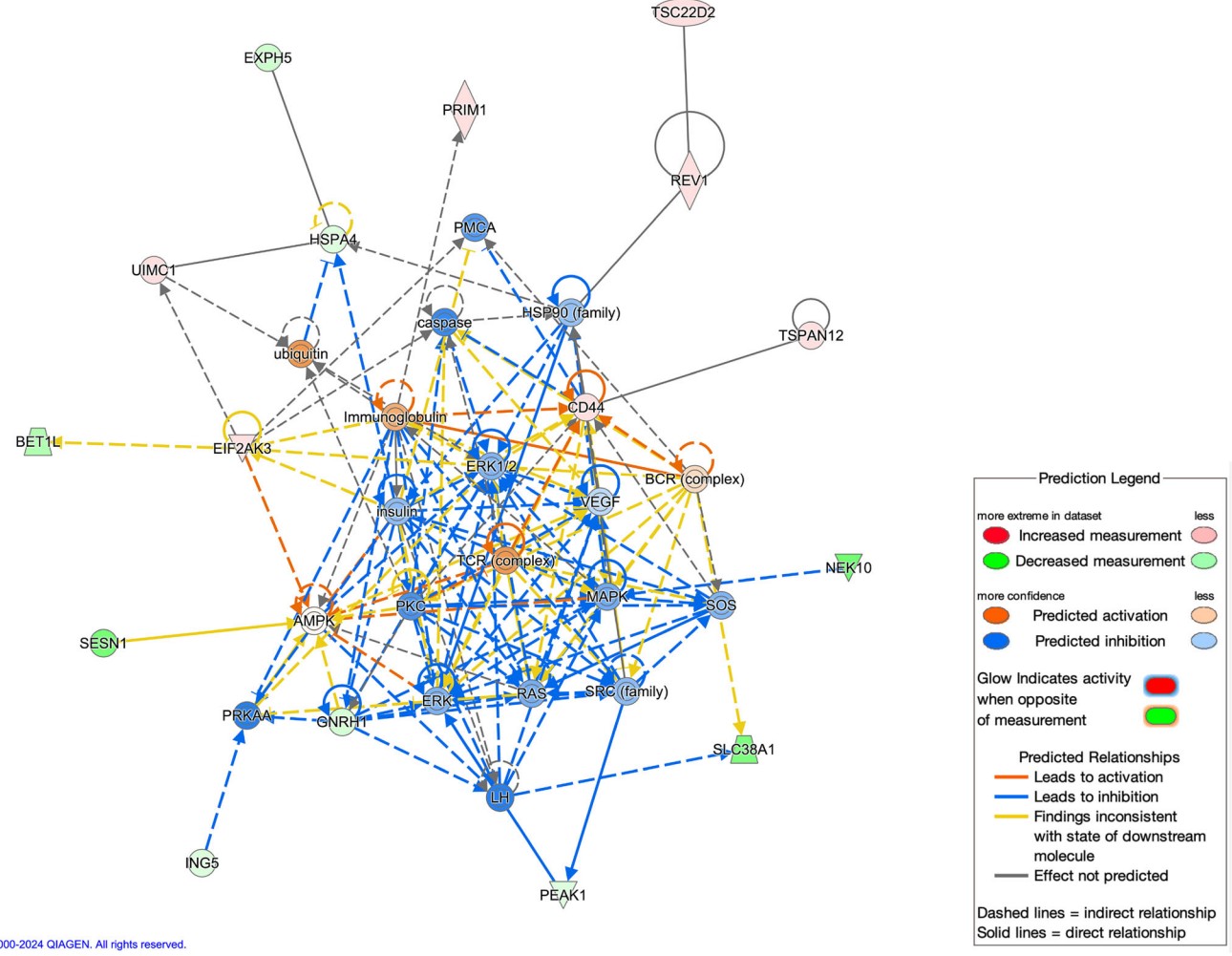

**Fig. 6 | The most enriched network from Ingenuity Pathway Analysis of significant colocalized genes was cancer, organismal injury and abnormalities, reproductive system disease.** Hypergeometric distribution testing with right-tailed Fisher's Exact Test; *p*-value threshold used (0.05).

fibroids, the African ancestry analysis was underpowered for certain analyses like pathway enrichment. However, we were able to detect both shared and unique risk genes. We used a combination of both race and genetically informed ancestry to conduct our population stratification due to limitations in the datasets available. We affirm that race and ancestry are not interchangeable, and race is not the ideal population descriptor in most genetic studies as it is a social construct[60]. Future data collection methods of large genomic studies should incorporate the use of genetic ancestry or other population descriptors like genetic similarity when stratifying data.

The GTEx samples used in our genetically predicted gene expression analyses come from European ancestry individuals, and while many genetic factors are shared across population groups, this limits the application to non-European ancestry analyses. There were only 129 samples included in the uterine tissue model[61], which affects our power to detect associations. Significant results in tissues such as testis and prostate can provide insight as the predicted gene expression may indicate relationships in reproductive and gonadal tissues. Increased representation of female-reproductive organs in tissue models would improve future discovery and validation. However, we were still able to identify at least one colocalized and significant predicted gene expression association in uterine tissue. Current studies are underway to add resources for gene expression of fibroids and the uterus, which can be utilized in future TWAS studies[62,63].

Heritability analysis can be subject to over-estimation when using SumHer[64], and we were limited in available annotation tags for the European and multi-ancestry analyses to the GBR HapMap3 annotation tags. However, as mentioned, our estimates were similar to previous reports that also used this approach and within the range of SNP-based heritability using other methods[31,34,38]. Our quality control LDSC analysis for the cross ancestry analysis was also limited in using European LD scores instead of a mixed ancestry option. Because European ancestry is the largest proportion of our multi-ancestry data set, we elected to use the European ancestry tags for this analysis. We were limited on further extrapolation of the gene-set enrichment tests as they only provide summarized correlation information, but the results could be utilized in future gene investigations.

In conclusion, this large-scale analysis identified several new shared and ancestry-specific genetic variations and genes of interest expressed within uterine tissue associated with fibroids. We detected known and novel genes in the under-researched African ancestry population and estimated a SNP-based heritability of 15.9%. This demonstrates the value of increasing and diversifying participation in GWAS. We have expanded the catalog of genetic associations with fibroids. Many of these genes have roles in DNA damage, cell cycle regulation, or are associated with fibroid risk factors like hypertension and vitamin D levels. These findings provide evidence for potential candidates for future research on fibroid etiology and targeted therapeutics.

## Methods

### Ethics

This research complies with all relevant ethical regulations, and was designated exempt non-human subjects research by the Vanderbilt University Medical Center Institutional Review Board as all study data was de-identified.

### Study population

We conducted a meta-analysis of summary statistics of female participants greater than 18 years old from previous GWAS of uterine fibroids (BioVU, eMERGE, Coronary Artery Risk Development in Young Adults [CARDIA], Black Women's Health Study [BWHS], United Kingdom Biobank [UKB], FinnGen, and Biobank Japan [BBJ])[20,24,27,33], and newly acquired results (BioVU, eMERGE, *All of Us* Research Program [*All of Us*]), described below. Each cohort uses one of several strategies to specify ancestral or demographic groups that we use as strata in our meta-analyses. The population descriptors of each data source and analysis filters are presented in Supplementary Data 1. Some of the phenotyping was based on survey response. Previous literature has shown that self-report can still be a useful approach, with one study validating 96% of self-reported diagnoses[65]. Genetic summary statistics from each meta-analysis are available in the GWAS Catalog (see Data Availability).

### BioVU

Our laboratory performed a genome-wide association study (GWAS) analysis using BioVU, Vanderbilt University's biorepository of DNA extracted from discarded blood collected during routine clinical testing and de-identified medical records in the Synthetic Derivative as previously described in ref. 66. Vanderbilt University Medical Center patients may sign the BioVU Consent Form to indicate whether they volunteer to donate their blood samples to this genome bank for studies on genomic and bioinformatics research on gene phenotypes and gene by environment interactions. As of January 2022, more than 275,000 DNA samples were recorded. BioVU fibroid phenotypes were defined by previously validated electronic medical record algorithm[67].

Newly acquired BioVU participant DNA samples were genotyped on a custom Illumina Multi-Ethnic Genotyping Array (MEGA-ex; Illumina Inc., San Diego, CA, USA). Quality control involved excluding samples or variants with missingness rates above 2%. Samples were also excluded if consent had been revoked, sample was duplicated, or sex concordance checks failed. Imputation was performed on the Michigan Imputation Server (MIS) v1.2.4 using Minimac4 and the Haplotype Reference Consortium (HRC) panel v1.1. The GWAS was performed using logistic regression implemented in SNPTEST (v2.5.4) in a race-stratified approach (non-Hispanic White, non-Hispanic Black) with covariates of age at fibroid diagnosis and first ten principal components (PCs).

### eMERGE

The eMERGE Network is a consortium of several EHR-linked biorepositories formed with the goal of developing approaches for the use of the EHR in genomic research. Consortium membership has evolved over eMERGE's 11-year history, with many sites contributing data including Group Health/University of Washington, Marshfield Clinic, Mayo Clinic, Northwestern University, Vanderbilt University, Children's Hospital of Philadelphia (CHOP), Boston Children's Hospital (BCH), Cincinnati Children's Hospital Medical Center (CCHMC), Geisinger Health System, Mount Sinai School of Medicine, Harvard University, and Columbia University. The eMERGE study was approved by the Institutional Review Board at each site and all methods were performed in accordance with the relevant guidelines and regulations. Cases were identified by presence of 2 or more ICD-9 (International Classification of Diseases) or ICD-10 codes for fibroids. A detailed description of the eMERGE network has been previously published[68].

Participants in the eMERGE network were genotyped separately, then imputed and merged. A detailed description of the genotyping, imputation, and quality control of the eMERGE phase III array dataset has been previously reported[69]. Summary statistics from eMERGE phases I-II were previously published and used our validated fibroid phenotype algorithm. Additional samples through phase III were phenotyped using phecodes, and race-stratified GWAS were performed with this data using logistic regression implemented in PLINK2 and adjusted for ten principal components of ancestry and study site.

**All of Us.** Limiting all participants in CDR V6 (C2022Q2R7) to female sex at birth, we used the SNOMED (Systematized Medical Nomenclature for Medicine -- Clinical Terminology) uterine fibroid code to find all females with condition codes for uterine fibroids. Age was determined by age at first condition. Females with two or more condition codes are considered a case. Females with one code had to be confirmed with at least one CPT (Current Procedural Terminology) code for pelvic imaging to be considered a case. We also used the Personal Medical History Survey for self-report. Females who selected "fibroids" to the following question "Has a doctor or health care provider ever told you that you have...? (select all that apply)" were also considered a case. Age for females with self-report data only was calculated as age at survey. Controls had to have at least two CPT codes for pelvic imaging with no history of fibroids, fibroid treatment, fibroid surgery, or self-reported fibroids. Age for controls was determined by age at 1st pelvic imaging. We filtered this cohort for females with whole genome sequencing. Lastly, we separated cases and controls into predicted ancestry groups in *All of Us*. Details about ancestry predictions can be found in the *All of Us* V6 WGS Quality Report. Quality control measures included minor allele frequency greater than 0.1% and call rate greater than 95%. The regression was adjusted for the top seven principal components and age as described above and was run using Hail Version-0.2.130. All uterine fibroid research was conducted in the *All of Us* research workbench. Details of the *All of Us* Research Program are previously reported[70,71].

### UK Biobank

The UKB is a large-scale biomedical database that aims to improve public health by enabling scientific discoveries. African, European, East Asian, and Central South Asian individuals were subset from the UKB. Fibroid status was defined using a combination of presence of 2 or more ICD-9 or ICD-10 codes or reporting of a fibroid diagnosis from survey response as defined previously[24]. We restricted to variants minor allele frequency greater than 0.5% and missingness less than 5%. Genetic analysis of imputed array data used logistic regression implemented in PLINK2 and adjusted for the first ten principal components.

### FinnGen

FinnGen is a large public-private partnership comprised of DNA and health data from up to 500,000 Finnish biobank participants[33]. We included published summary statistics from FinnGen release 8 for uterine fibroids which used ICD-10 diagnostic codes. Their methods are available on their online documentation, https://finngen.gitbook.io/. In brief, their association tests are run using regenie, adjusted for age, ten principal components, and genotyping batch as covariates. They used the approximate Firth test for variants with an initial *p*-value less than 0.01[33].

### Biobank Japan

Biobank Japan is a multi-institutional hospital-based registry comprised of DNA and medical records from individuals of Japanese ancestry. We utilized published summary statistics for uterine fibroids identified with diagnostic codes phecode 218.1. These methods are publicly available, and the binary trait association test included

adjustments for age, age[2], and the top 20 principal components using a generalized linear mixed model in SAIGE (v.0.37)[27].

### Black Women's Health Study
Black Women's Health Study (BWHS) is a U.S. prospective cohort study of 59,000 United States Black women who enrolled in the study in 1995 and have been followed by biennial questionnaires since then. Uterine fibroid diagnoses were ascertained by self-report from premenopausal participants, with medical record validation among a subset of cases[65,72,73]. GWAS of fibroids in BWHS was previously described. Briefly, the summary statistics were generated from a SNPTEST2 single-variant association analysis adjusted for the first ten principal components[20].

### CARDIA
CARDIA is a prospective multi-center study with 5115 adult White and Black participants of the age group 18 to 30 years, recruited from four centers. Details of the CARDIA study design have been previously published[74]. The CARDIA Women's Study performed standardized study ultrasounds on women from CARDIA to detect the presence or absence of fibroids[75]. The fibroids GWAS has been previously described; briefly GWAS was performed using SNPTEST2 and adjusted for the first five principal components[20].

### Meta-analysis
Four fixed-effect inverse variance-weighted meta-analyses were performed using METAL software v(2011-3-25)[76]: the groups being genome-wide summary statistics of European ancestry, African ancestry, East Asian/Central South Asian ancestry, and all data combined (multi-ancestry). Stratification into analysis groups was based on either genetically determined ancestry or self-reported/electronic health record race due to data availability limitations (Supplementary Data 1).

Statistical significance was at the standard genome wide significance level of $p < 5 \times 10^{-8}$ using Bonferroni correction. The genomic inflation factor was calculated for each meta-analysis. The LDSC intercept was estimated using LD score regression (v1.0.1) for each meta-analysis group with common variants from HapMap3[35]. Each variant had to be present in at least two data sets in each meta-analysis to be included in subsequent analyses. Variants were mapped to genes based on location and OpenTarget Genetics[77]. The correlation of effect sizes between meta-analyses was performed using R function corr.

### SNP-based heritability
SNP-based heritability ($h^2_{SNP}$) of fibroids was also estimated using SumHer[34] version 5.2. We used the pre-computed UK Biobank based BLD-LDAK model (4 models). We set the check-sums flag to NO to ignore the fact that some markers were not included in the annotated tag list, but only 0.14% were at most not overlapped. Common variants restricted to HapMap3 SNPs were used to estimate heritability. Insertions and deletions were excluded from SumHer as SumHer only accepts single nucleotide variations.

### FUMA
Functional gene annotation was done using the online Functional Mapping and Annotation (FUMA)[36] tool for all four meta-analyses summary statistics (accessed August 2023). FUMA identified sentinel variants determined as variants in low LD ($r^2 < 0.1$). ANNOVAR annotations are performed for all variants, and their Gene2Func application provided differentially expressed gene and gene-set enrichment analyses[36]. Variants are mapped to genes and those are used for enrichment tests like differentially expressed gene sets and Gene Ontology sets. Regional locus-zoom plots were generated using FUMA.

### Conditional analysis and variant categorization
We used Genome-wide Complex Trait Analysis (GCTA)[78] version 1.93.0 to perform conditional-joint association testing for secondary signals. We used our FUMA-determined sentinel SNPs to identify secondary signals. Then we used the GWAS-catalog SNPs associated with fibroids to identify secondary signals to previously discovered variants. We used public databases like OpenTarget Genetics[79] and the GWAS catalog to determine if these were novel gene associations, previously unpublished gene associations, or secondary signals in known gene associations.

### Genetically predicted gene expression
We used S-PrediXcan (v0.7.1) with JTI models to conduct genetically predicted gene expression[80,81]. S-PrediXcan conducts association tests between phenotypes and gene expression levels predicted by the genetic variants provided as summary statistics based on a library of tissues from the Genotype-Tissue Expression (GTEx) project. We used covariance matrices built for all 49 GTEx v8 tissues. European genetic based covariance matrices were used for all four analysis groups. We conducted S-PrediXcan with each of the four meta-analysis summary statistics. Significance cut off was at $9.4 \times 10^{-8}$ which was determined on the number of gene-tissue pairs tested. Further filtering of results was conducted based on colocalization analyses.

### Colocalization
We performed colocalization analysis between the summary statistics from the meta-analyses and gene expression in GTEx v8 models in 49 tissues. The Approximate Bayes Factor (ABF) analyses were calculated using the R library package coloc (v5.2.2) and the method coloc.abf with the default priors. Significant colocalized genes were identified as those with a hypothesis 4 (H4) probability greater than 0.8, hypothesis 3 (H3) probability less than 0.3, and GPGE $p$-value less than the Bonferroni corrected significance level of $9.4 \times 10^{-8}$. H3 is the hypothesis that both traits are associated and have different single causal variants. H4 is the hypothesis that both traits are associated and share the same single causal variant[82].

### Gene pathway analysis
The combined list of significant and colocalized gene-tissue results from S-PrediXcan (results from all four meta-analysis) were analyzed using the core analysis function in Ingenuity Pathway Analysis (IPA) software (Qiagen, accessed March 2024). IPA software algorithms employed have been described previously. All analyses including canonical signaling pathways, diseases and functions, and regulatory networks were ordered by enrichment $p$-value[83].

### Reporting summary
Further information on research design is available in the Nature Portfolio Reporting Summary linked to this article.

## Data availability
Genetic association summary statistics for each meta-analysis have been deposited in the GWAS Catalog under accession codes GCST90461957 (multi-ancestry), GCST90461958 (European ancestry), GCST90461959 (East Asian/Central South Asian ancestry), GCST90461960 (African ancestry). Study-specific summary statistics for BioBank Japan are available on https://www.pheweb.jp/pheno/UF, and those for FinnGen are available on https://r7.finngen.fi/pheno/CD2_BENIGN_LEIOMYOMA_UTERI. UKBB data access can be requested from https://www.ukbiobank.ac.uk/enable-your-research/apply-for-access. BioVU [https://victr.vumc.org/how-to-use-biovu/] and eMERGE [https://emerge-network.org/collaborate/] data also require approved access, which can be requested at their respective links. The All of Us [https://www.researchallofus.org/] data is accessible on the Researcher Workbench with registered access. The data from CARDIA have been deposited at

dbGaP accession phs000285.v3.p2 [https://www.ncbi.nlm.nih.gov/projects/gap/cgi-bin/study.cgi?study_id=phs000285.v3.p2] and the BWHS data at dbGaP accession phs001409.v2 [https://www.ncbi.nlm.nih.gov/projects/gap/cgi-bin/study.cgi?study_id=phs001409].The predicted expression models used are publicly available on https://predictdb.org/post/2021/07/21/gtex-v8-models-on-eqtl-and-sqtl/. The other data generated in this study are provided in the Supplementary Data files.

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

## Acknowledgements

This work was supported in by UL1TR000445 (JK), K12AR084232 (JNH, EAJ, MMS: PI: DRVE), R01HD074711(DRVE), R03HD078567(DRVE),

R01HD093671 (DRVE/TLE), and R01HD112169 (DRVE/TLE). Additional support was provided by R01GM139891 (WW), K25DK128563 (AK), ZIAHG200417 (AW, JMK: PI: JCD), U01CA164974 (JRP). Uterine fibroid-related work from the Black Women's Health Study (BWHS) was supported in part by R01HD057966 (LAW), R01CA098663 (JRP). Vanderbilt University Medical Center's BioVU projects are supported by numerous sources: institutional funding, private agencies, and federal grants. These include NIH funded Shared Instrumentation Grant S10OD017985, S10RR025141, and S10OD025092; CTSA grants UL1TR002243, UL1TR000445, and UL1RR024975. Genomic data are also supported by investigator-led projects that include U01HG004798, R01NS032830, RC2GM092618, P50GM115305, U01HG006378, U19HL065962, R01HD074711; and additional funding sources listed here: https://victr.vumc.org/biovu-funding/. The research has been conducted using the UK Biobank Resource (Application #13869) http://www.ukbiobank.ac.uk. We want to acknowledge the participants and investigators of the FinnGen study. This work is supported by multiple eMERGE phases. Phase 1: The eMERGE Network was initiated and funded by NHGRI, in conjunction with additional funding from NIGMS through the following grants: U01HG004610 (Group Health Cooperative/University of Washington); U01HG004608 (Marshfield Clinic Research Foundation and Vanderbilt University Medical Center); U01HG04599 (Mayo Clinic); U01HG004609 (Northwestern University); U01HG04603 (Vanderbilt University Medical Center, also serving as the Administrative Coordinating Center); U01HG004438 (CIDR) and U01HG004424 (the Broad Institute) serving as Genotyping Centers. Phase 2: The eMERGE Network was initiated and funded by NHGRI through the following grants: U01HG006389 (Essentia Institute of Rural Health, Marshfield Clinic Research Foundation and Pennsylvania State University); U01HG006382 (Geisinger Clinic); U01HG006375 (Group Health Cooperative/University of Washington); U01HG006379 (Mayo Clinic); U01HG006380 (Icahn School of Medicine at Mount Sinai); U01HG006388 (Northwestern University); U01HG006378 (Vanderbilt University Medical Center); and U01HG006385 (Vanderbilt University Medical Center serving as the Coordinating Center); U01HG004438 (CIDR) and U01HG004424 (the Broad Institute) serving as Genotyping Centers. Phase 3: This phase of the eMERGE Network was initiated and funded by the NHGRI through the following grants: U01HG008657 (Group Health Cooperative/University of Washington); U01HG008685 (Brigham and Women's Hospital); U01HG008672 (Vanderbilt University Medical Center); U01HG008666 (Cincinnati Children's Hospital Medical Center); U01HG006379 (Mayo Clinic); U01HG008679 (Geisinger Clinic); U01HG008680 (Columbia University Health Sciences); U01HG008684 (Children's Hospital of Philadelphia); U01HG008673 (Northwestern University); U01HG008701 (Vanderbilt University Medical Center serving as the Coordinating Center); U01HG008676 (Partners Healthcare/Broad Institute); U01HG008664 (Baylor College of Medicine); and U54MD007593 (Meharry Medical College).

## Author contributions

Conceptualization: J.K., T.L.E., D.R.V.E., J.N.H.; Data curation: J.K., A.W., S.H.J., L.A.W., T.L.E., D.R.V.E., J.N.H.; Formal analysis: J.K., A.W., H.N., J.J., M.M.S.; Funding acquisition T.L.E., D.R.V.E.; Investigation: J.K., H.N., M.M.S.; Project administration: S.H.J., T.L.E., D.R.V.E., J.N.H.; Resources: J.M.K., M.A.A., J.R.P., L.J.R.T., L.C.K., E.A.R.N., L.A.W., J.C., J.C.D., W.Q.W.; Supervision: T.L.E., D.R.V.E., J.N.H.; Visualization: J.K., H.N.; Writing – original draft: J.K., T.L.E., D.R.V.E., J.N.H.; Writing – review and editing: J.K., A.W., M.M.S., L.A.W., A.K., E.A.J., E.A.R.N., L.J.R.T., L.C.K., J.R.P., M.A.A., J.C.D., P.M., K.K., W.Q.W.

## Competing interests

The authors declare no competing interests.

## Additional information

[1]Division of Quantitative and Clinical Sciences, Department of Obstetrics & Gynecology, Vanderbilt University Medical Center, Nashville, TN, USA. [2]Medical Scientist Training Program, Vanderbilt University School of Medicine, Nashville, TN, USA. [3]Division of Genetic Medicine, Department of Medicine, Vanderbilt University Medical Center, Nashville, TN, USA. [4]Vanderbilt Genetics Institute, Vanderbilt University, Nashville, TN, USA. [5]Center for Precision Health Research, National Human Genome Research Institute, National Institute of Health, Bethesda, MD, USA. [6]Tufts University Medical School Graduate Programs, Boston, MA, USA. [7]Medicine, Health and Society, Vanderbilt University, Nashville, TN, USA. [8]Department of Biomedical Informatics, Vanderbilt University Medical Center, Nashville, TN, USA. [9]Institute for Medicine and Public Health, Vanderbilt University Medical Center, Nashville, TN, USA. [10]Division of Epidemiology, Department of Medicine, Vanderbilt Genetics Institute, Vanderbilt University Medical Center, Nashville, TN, USA. [11]Department of Nutritional Sciences University of Michigan School of Public Health, Ann Arbor, MI, USA. [12]Department of Epidemiology, Boston University School of Public Health, Boston, MA, USA. [13]Slone Epidemiology Center at Boston University, Boston, MA, USA. [14]Center for Applied Genomics, Department of Pediatrics, Children's Hospital of Philadelphia, Philadelphia, PA, USA. [15]All of Us Research Program, Office of the Director, National Institutes of Health, Bethesda, MD, USA. [16]Division of Nephrology, Department of Medicine, Vagelos College of Physicians & Surgeons, Columbia University, New York, NY, USA. [17]Northwestern University Feinberg School of Medicine, Chicago, IL, USA. [18]Preventive Medicine, Northwestern University Feinberg School of Medicine, Chicago, IL, USA. [19]Center for Autoimmune Genomics and Etiology, Department of Pediatrics, Cincinnati Children's Hospital, University of Cincinnati, Cincinnati, OH, USA. [20]University of

Arizona Comprehensive Cancer Center, Tucson, AZ, USA. [21]Department of Health Promotion Sciences, Mel & Enid Zuckerman College of Public Health, University of Arizona, Tucson, AZ, USA. [22]Public Health Research Institute of India, Mysuru, India. [23]Department of Medicine, College of Medicine, University of Arizona, Tucson, AZ, USA. [24]Public Health Practice, Policy, & Translational Research Department, Mel & Enid Zuckerman College of Public Health, Phoenix, AZ, USA. [25]Vanderbilt Epidemiology Center, Vanderbilt University Medical Center, Nashville, TN, USA. [26]These authors contributed equally: Todd L. Edwards, Digna R. Velez Edwards, Jacklyn N. Hellwege. [27]These authors jointly supervised this work: Digna R. Velez Edwards, Jacklyn N. Hellwege.
✉e-mail: digna.r.velez.edwards@vumc.org; jacklyn.hellwege@vumc.org

