## [Peer Review File · Nature Communications]

REVIEWER COMMENTS

Reviewer #1 (Remarks to the Author):

The authors present results from a multi-ancestry GWAS meta-analysis of uterine fibroids of which 28% of cases and 18% of controls are of non-European descent. The authors then characterize significant genes into functional pathways and enrichment sets. I have several comments on the methodology:

1. The authors mentioned using self-reported history of fibroids as a method for determining a case from several of the databases such as the All of Us, and with no validation even among a subset of cases. What is the accuracy of self-reported fibroids? What is the specificity and sensitivity of the self-reported personal history data. Additionally, in the All of Us, the age for females with self-reported data only was taken as the age of survey, which can bias results, as this data is combined with females where their age is taken as age of uterine fibroids code.

2. The threshold for significant colocized genes were identified as those with a PP.H4 greater than 0.6, which is very lenient. A threshold of 0.8 is more commonly used. Suggest changing this to the more commonly used $PP.H4 > 0.8$.

3. The authors categorize novel associations as genes that have never been reported before and "previously unpublished" genes as those that have been reported in biobanks but not in literature. Did the authors filter by previously reported genes that may be in LD with their novel genes? Suggest checking if any of the 59 novel genes are in LD with any previously reported genes.

4. Can the authors provide any elaboration or hypothesis on why refractive error and prostate cancer would come up in the enriched gene sets?

5. There is no external replication of any of the findings.

Reviewer #2 (Remarks to the Author):

Overview: Kim et.al performed a large-scale ancestry-specific and multi-ancestry (European, Asian, and African) meta-analysis of genome-wide association studies for women with and without Uterine fibroids. Their analysis is an important addition to understand genetic risk of Uterine fibroids for individuals with non-European ancestry that has been underrepresented in previous studies. Briefly, the authors identified 11 novel fibroid-associated genes through meta-analysis and 59 novel genes through TWAS and colocalization analysis. This manuscript is written well, and results presented clearly. With that said I have several comments.

Major Comments:

1. To better show effect size correlation between ancestry groups, can the authors perform a regression on SNP effect size between ancestry group, weighted by standard error (SE) of effect estimate and adjust for MAF as covariate? For example, fit a weighted regression of effect size(EUR) \sim effect size(AFR) + MAF(AFR), using SE from AFR as weight. This approach accounts for uncertainty in estimates and MAF difference between ancestry group.
2. Can the authors provide more details in the method section on how LDSC, SumHer and S-PrediXcan is implemented? For example, did the authors filter out rare genetic variants? Did the authors filter genes by cross-validation R2 for TWAS in S-PrediXcan?
3. Can the authors expand discussion regarding heritability estimates in multi-ancestry meta-analysis? For example, the multi-ancestry results depend on a combination of LD patterns, and how to interpret this in the context of EUR only LD scores.
4. It would be helpful if the authors provide discussion on how GO terms may be differentially enriched across ancestry groups solely as a function of power, and less of biology. This difference can be driven by differential power at each SNP site (map to different genes) due to different LD structure, allele frequency and environmental factor across ancestry, i.e., not reflecting that the biological pathway associated with Uterine fibroids is different across ancestry.

Minor Comments:

1. When the authors mentioned a SNP mapped to a gene, e.g, line 142-145, please make it clear if it's an exon variant, an intron variant mapped to the closest gene, or mapped to a gene by any other means.
2. Add figure number and description in caption for each figure in the supplemental material.
3. Check odds ratio, CI and p values reported in line 164-165. These values are identical to values reported in line 143.
4. Change “de-enrichment” to “depletion” in line 212.

Reviewer #3 (Remarks to the Author):

Major Comments:

In “Genome-wide meta-analysis identifies novel risk loci for uterine fibroids within and across multiple ancestries,” the authors perform a large-scale, multi-ancestry meta-analysis of genome-wide association studies conducted across more than 500,000 female cases and controls of uterine fibroids. At a high-level, the authors identified 370 independently associated signals that included replications of well-established uterine fibroid loci (e.g., TP53, SYNE1) as well as 2 novel and 6 previously unpublished loci. SNP-based heritability was estimated in the meta-, European-specific, East Asian-specific, and African ancestry-specific analyses to be 0.05, 0.07, 0.115, and 0.15, respectively. To the best of my knowledge, the GWAS would represent the largest one published to date on uterine fibroids.

The work is of outstanding quality and generates a novel set of summary statistics that can be used as input for numerous downstream research and drug development analyses. Functional annotation, gene set enrichment analyses, transcriptome-wide association studies, and ingenuity pathway analyses leveraging the meta-analysis and ancestry-specific summary statistics offer novel insight into uterine fibroid biology. Several points of concern however must be addressed.

One area of focus that would substantially strengthen the paper is the addition of cross-trait analyses. For example, one possible analysis could investigate whether any novel phenotypes are now observed to have significant overlapping heritability with a uterine fibroids GWAS with greater statistical power. Another could potentially leverage Mendelian randomization to perform causal inferences to interrogate relationships with factors that have been associated with fibroids as mentioned on page 5 between lines 113 and 117.

One observation that gives pause is the low degree of correlation ($r^2 = 0.05$) observed between beta estimates obtained in GWAS performed in analyses of individuals that are all of either African or European ancestry (Supplemental Figure 2a). The authors attribute the anomaly to small sample size in the discussion, but I’m curious to know whether that interpretation accounts for the observation of African ancestry-specific analyses showing a higher degree of SNP-based heritability than European ancestry? Additionally, what insight, if any, does the observation of greater heritability in African ancestry provide as to why that population is affected with a greater prevalence of fibroids and more severe fibroids?

The authors do not discuss much about the heterogeneity observed in the meta-analyses. Could the authors include the degree to which the significant observations were also significantly

heterogeneous across the cohorts? It would also be useful to incorporate comments on the discussion about sources of heterogeneity. Are there specific cohorts that are routinely showing outlier behavior for effect size estimates?

The GWAS as described in the methods were “performed using logistic regression... .. adjusting for 10 principal components of ancestry.” No mention is made about adjustments for age or BMI. Were these omitted for any particular reason? Most previous GWAS on uterine fibroids include adjustments for these covariates. Furthermore, in the text of the methods, the information regarding the regression model used for the cohort is not provided for All of Us, UK Biobank, FinnGen, or Biobank Japan. Please add these to the text. The models used in the GWAS of fibroids for BWHS and CARDIA are provided in citations, though it would be a worthwhile consideration to briefly include a description to the reader.

Figure 1 provides a great overview of the different analyses. It would be useful to the reader to have a sense of what fractions of the cases and controls for each analysis come from each biobank included. For example, perhaps in a revised version of the figure, the authors could replace the graphics representing each cohort (i.e., “BioVU DNA Repository”) with a small table outlining the number of cases and controls coming from each individual cohort. Alternatively, the data could be organized and referenced in a supplemental table or figure.

Minor Comments and Suggestions:

Page 4, line 78: Adjust text to “...highly heritable, common, and benign tumors of the uterus with poorly understood etiology.”

Page 5, line 106: Remove commas and adjust to “...cumulative incidence of fibroids by age...”

Page 5, line 115: Consider “self-selected” in place of “self-identified”

Page 7, line 156: Indicate “Supplemental Figure 1a”

Page 11, line 276: Make “Supplemental Table 7b” bold text

Reviewer #4 (Remarks to the Author):

Response to Reviewer Feedback

Reviewer #1

1. *The authors mentioned using self-reported history of fibroids as a method for determining a case from several of the databases such as the All of Us, and with no validation even among a subset of cases. What is the accuracy of self-reported fibroids? What is the specificity and sensitivity of the self-reported personal history data. Additionally, in the All of Us, the age for females with self-reported data only was taken as the age of survey, which can bias results, as this data is combined with females where their age is taken as age of uterine fibroids code. (Reviewer 1)*

We acknowledge that self-reported history of fibroids is not the most ideal case determination approach. Most of our data sources use medical diagnosis or imaging to confirm fibroid status, however, not all sources have this available. For instance, in All of Us, our phenotyping approach used a combination of medical diagnosis and self-reported fibroids (47% of cases). We are unable to validate the All of Us cases that are based on survey response due to the inability to chart review in this databank. Previous literature (Wise, et al 2007, PMID: 15738025) has shown that self-report can still be a useful approach, and in their study (BWHS) they validated 96% of self-reported diagnoses. Additionally, it is more likely that self-report would misclassify controls rather than cases as they may have asymptomatic or undiagnosed fibroids. This would, however, bias our results towards the null rather than increase risk of false positive findings. We agree that the age variable for All of Us is not informative as to the age at which fibroids originated, however age at diagnosis is not age at initial development of fibroids, as this is usually not known. Therefore, age covariates are imprecise and reflect only the ages at which fibroids were reported.

2. *The threshold for significant colocalized genes were identified as those with a PP.H4 greater than 0.6, which is very lenient. A threshold of 0.8 is more commonly used. Suggest changing this to the more commonly used PP.H4 > 0.8. (Reviewer 1)*

Given the use of Bayesian statistics, thresholds for posterior probabilities are generally not standardized. We have adjusted our results (text, tables) to reflect the requested threshold of 0.8, along with subsequent analyses such as IPA to use this updated filtered input list.

3. *The authors categorize novel associations as genes that have never been reported before and "previously unpublished" genes as those that have been reported in biobanks but not in literature. Did the authors filter by previously reported genes that may be in LD with their novel genes? Suggest checking if any of the 59 novel genes are in LD with any previously reported genes. (Reviewer 1)*

We performed a conditional analysis accounting for all previously reported SNPs in the GWAS catalog to identify whether our signals were independent of or in LD with any previously known associations. We utilized the list of conditionally independent SNPs to then map to genes. We did not directly evaluate LD across entire gene regions after determining which gene(s) mapped to the independent SNP signals as this is complex. We have included regional plots for all novel and unpublished in previous literature sentinel SNPs in the supplemental figures (Supplemental Figures 6A-K).

4. *Can the authors provide any elaboration or hypothesis on why refractive error and prostate cancer would come up in the enriched gene sets? (Reviewer 1)*

FUMA maps SNPs to genes based on factors like ANNOVAR annotation and distance to genes. MAGMA is used to compute a gene-based p-value. For the gene-set enrichment tests, the gene set p-value is computed based on the gene-based p-values for various gene sets. The gene sets the reviewer is referring to is specific to genes previously implicated from genome-wide association studies. It is possible that the indicated genes may not be the true causal gene effecting the phenotype. Among the 37 genes that overlapped between fibroids and prostate cancer, 17 of them are associated with other tumor pathologies as well, likely contributing to the observed enrichment. Steroid hormone regulation is also a common theme across these conditions. Refractive error is very polygenic, with 1381 genes included in the GWAS catalog set, so it is difficult to interpret 94 overlapping genes. When we put these 94 genes into FUMA's Gene2Func analysis, there seems to be an enrichment in SUMOylation processes which is broadly related to protein regulation. We limit our interpretation of the gene-set enrichment tests as they only provide correlation information on a summarized level, and rather provides potential genes to further investigate rather than hypothesize on from just this test. We have added text describing this limitation to the main text in the discussion.

5. *There is no external replication of any of the findings. (Reviewer 1)*

While there is no external replication in this study design, due to the limitations of available independent data outside of the sources used for this large meta-analysis, we were able to replicate hundreds of previously identified variant associations with fibroids in the GWAS catalog, supporting the validity of our findings. Previous research has shown single joint analysis rather a two-stage replication analysis confers increased power to detect associations despite stringent significance levels (Skol, et. al, 2006, PMID: 16415888).

Reviewer #2

1. *To better show effect size correlation between ancestry groups, can the authors perform a regression on SNP effect size between ancestry group, weighted by standard error (SE) of effect estimate and adjust for MAF as covariate? For example, fit a weighted regression of effect size(EUR) ~ effect size(AFR) + MAF(AFR), using SE from AFR as weight. This approach accounts for uncertainty in estimates and MAF difference between ancestry group. (Reviewer 2)*

Variants that should have been filtered out (those present in only one dataset) were included in parts of the analysis such as the effect correlation, and we have re-plotted the effect size comparisons restricted to these SNPs and plotted effect sizes without recoding to the European effects being all positive. The new correlation is much higher (0.44) and our plots have been updated accordingly. All results have been updated to ensure this filter was applied for all analyses.

2. *Can the authors provide more details in the method section on how LDSC, SumHer and S-PrediXcan is implemented? For example, did the authors filter out rare genetic*

variants? Did the authors filter genes by cross-validation R² for TWAS in S-PrediXcan? (Reviewer 2)

The JTI models used for S-PrediXcan did not include rare genetic variants (MAF<0.05). We did not filter genes by cross-validation R², instead the models were only made available if q-values were less than 0.05 (Barbeira, et.al 2018, PMID: 29739930, Zhou et.al 2021, PMID: 33020666). In order to be as stringent as possible, we further restricted our subsequent analyses such as IPA to those genes which also colocalized. For both LDSC and SumHer, we used variants restricted to HapMap3 variants which include common variants. We have added additional details to the methods section. The minor allele frequency thresholds have been added as a new column to the Supplemental Table 1.

- 3. Can the authors expand discussion regarding heritability estimates in multi-ancestry meta-analysis? For example, the multi-ancestry results depend on a combination of LD patterns, and how to interpret this in the context of EUR only LD scores. (Reviewer 2)*

We agree with the reviewer that LD scores affect our ability to conduct heritability estimates. This is a limitation of available data as LD scores from a mixed sample would be difficult to interpret. Because European ancestry is the largest proportion of our multi-ancestry data set, we elected to use the European tags for the multi-ancestry summary statistics. However, due to this inherent limitation, we have also performed and provide ancestry-specific heritability estimates using ancestry-matched LD statistics from continental reference populations which are more interpretable. The multi-ancestry estimates do not deviate greatly from the single-ancestry estimates, likely reflecting the ancestral composition of the overall set. We have added this consideration of multi-ancestry LD to our discussion section.

- 4. It would be helpful if the authors provide discussion on how GO terms may be differentially enriched across ancestry groups solely as a function of power, and less of biology. This difference can be driven by differential power at each SNP site (map to different genes) due to different LD structure, allele frequency and environmental factor across ancestry, i.e., not reflecting that the biological pathway associated with Uterine fibroids is different across ancestry. (Reviewer 2)*

We agree with the reviewer that the differences observed in this gene set enrichment analysis could largely be due to differences in power to detect rather than biological differences in fibroids pathophysiology between population groups. This analysis provides a way to interpret the differentially powered results and the collective and combination of results may be more informative rather than comparing the results. We have added text to better highlight this point, such as “differentially enriched across ancestry groups as a function of power, and less of biology.”

- 5. When the authors mentioned a SNP mapped to a gene, e.g, line 142-145, please make it clear if it's an exon variant, an intron variant mapped to the closest gene, or mapped to a gene by any other means.*

Variants described in line 142-145 were mapped to nearest genes by distance to transcription start sites. The variant type is noted and if the variant is not physically located in the gene body, nearest gene is used.

6. *Add figure number and description in caption for each figure in the supplemental material.*

We have added figure numbers and descriptions in captions for each figure in the supplemental material.

7. *Check odds ratio, CI and p values reported in line 164-165. These values are identical to values reported in line 143.*

Thank you for your attention to detail. In fact, rs78378222 did have the same OR, CI, and P-value in both cross ancestry and European analyses as all data contributing to the estimate was from European ancestry datasets only. We have added text to highlight this reason. For SNP rs58415480 the OR and CI are the same due to rounding, but the p-values are different.

8. *Change “de-enrichment” to “depletion” in line 212.*

This change has been made.

Reviewer #3

1. *One area of focus that would substantially strengthen the paper is the addition of cross-trait analyses. For example, one possible analysis could investigate whether any novel phenotypes are now observed to have significant overlapping heritability with a uterine fibroids GWAS with greater statistical power. Another could potentially leverage Mendelian randomization to perform causal inferences to interrogate relationships with factors that have been associated with fibroids as mentioned on page 5 between lines 113 and 117. (Reviewer 3)*

We agree with the reviewer that additional cross-trait analyses would strongly contribute to the field. We currently have manuscripts accepted from our lab that have performed a comprehensive cross-trait approach using Mendelian randomization and genetic correlation methods. These works also have been able to subset the data used in the fibroid meta-analysis to have ample independent data sets to use in these other cross-trait analyses, which places them somewhat outside the scope of this manuscript.

2. *One observation that gives pause is the low degree of correlation ($r^2 = 0.05$) observed between beta estimates obtained in GWAS performed in analyses of individuals that are all of either African or European ancestry (Supplemental Figure 2a). The authors attribute the anomaly to small sample size in the discussion, but I'm curious to know whether that interpretation accounts for the observation of African ancestry-specific analyses showing a higher degree of SNP-based heritability than European ancestry? Additionally, what insight, if any, does the observation of greater heritability in African ancestry provide as to why that population is affected with a greater prevalence of fibroids and more severe fibroids? (Reviewer 3)*

We have re-made the effect correlation plots as mentioned above (Reviewer 2 comment 1). The correlation is much higher ($R = 0.44$) and was being largely influenced by a few outliers in our previous plots that had not been filtered out appropriately to have been present in more than one dataset in both analysis groups being compared.

The higher degree of SNP-based heritability could be due to 1) the effects at the causal loci are larger in the African ancestry group, 2) there are additional distinct causal loci in these African ancestry data that do not exist in the other data sets, 3) the allele frequency of causal alleles are systematically higher in our African ancestry group, and 4) a combination of the above.

Regarding the first point, this would imply effect modification by ancestry, which may be due to gene-environment interactions for factors that are experienced more commonly in that group compared to others. Both 2 and 3 imply selection, and previous work from our lab have reported evidence of higher allele frequencies at loci associated with fibroproliferative traits in African populations (Hellwege, et. al 2017, PMID: 28792542).

3. *The authors do not discuss much about the heterogeneity observed in the meta-analyses. Could the authors include the degree to which the significant observations were also significantly heterogeneous across the cohorts? It would also be useful to incorporate comments on the discussion about sources of heterogeneity. Are there specific cohorts that are routinely showing outlier behavior for effect size estimates? (Reviewer 3)*

We have included the heterogeneity p-value in the supplemental tables and the I-squared value to provide additional information about the heterogeneity observed (Supplemental Tables 2 and 3). In the main text, we have also edited the results section to highlight the variants with a heterogeneity p-value < 0.01 in our novel/previously unpublished variants (Supplemental Table 3). We have plotted the effect sizes for each cohort for the novel, previously unpublished, and secondary single variants (Supplemental Table 3 variants) in the graphs below. The x-axis is each data cohort used in the meta-analysis and the y-axis represents the beta effect estimate. We noted no routine pattern of cohorts showing outlier behavior for effect size estimates, although there are individual outliers.

Effect sizes for Independent and Novel SNPs for each cohort in the cross ancestry analysis

AOU AFR	1
AOU EUR	2
BBJ EAS	3
BioME AFR	4
BioVU AFR	5
BioVU EUR	6
BioVU MEGA AFR	7
BioVU MEGA EUR	8
BWHS AFR	9
CARDIA AFR	10
CARDIA EUR	11
CHOP EUR	12
eMERGE 123 AFR	13
eMERGE 123 EUR	14
eMERGE 1 AFR	15
eMERGE 1 EUR	16
FinnGen EUR	17
Geisinger EUR	18
Mayo EUR	19
MtSinai AFR	20
NW EUR	21
UKBB CSA	22
UKBB EAS	23
UKBB AFR	24
UKBB EUR	25

Effect sizes for Independent and Novel SNPs for each cohort in the European ancestry analysis

AOU EUR	1
BioVU EUR	2
BioVU MEGA EUR	3
CARDIA EUR	4
CHOP EUR	5
eMERGE 123 EUR	6
eMERGE 1 EUR	7
FinnGen EUR	8
Geisinger EUR	9
Mayo EUR	10
NW EUR	11
UKBB EUR	12

4. *The GWAS as described in the methods were “performed using logistic regression... adjusting for 10 principal components of ancestry.” No mention is made about adjustments for age or BMI. Were these omitted for any particular reason? Most previous GWAS on uterine fibroids include adjustments for these covariates. Furthermore, in the text of the methods, the information regarding the regression model used for the cohort is not provided for All of Us, UK Biobank, FinnGen, or Biobank Japan. Please add these to the text. The models used in the GWAS of fibroids for BWHS and CARDIA are provided in citations, though it would be a worthwhile consideration to briefly include a description to the reader. (Reviewer 3)*

Each GWAS was performed differently as most of the data were provided as summary statistics from previously conducted GWAS. In the majority of the datasets, age or BMI were not available, and some GWAS did include these covariates. Our previous unpublished work has shown that age and BMI do not substantively modify SNP effect sizes. We have added additional information regarding the regression models for each dataset in the text of the methods.

5. *Figure 1 provides a great overview of the different analyses. It would be useful to the reader to have a sense of what fractions of the cases and controls for each analysis come from each biobank included. For example, perhaps in a revised version of the*

figure, the authors could replace the graphics representing each cohort (i.e., “BioVU DNA Repository”) with a small table outlining the number of cases and controls coming from each individual cohort. Alternatively, the data could be organized and referenced in a supplemental table or figure. (Reviewer 3)

The case and control information for each cohort, along with additional demographic data, is shown in Supplemental Table 1. We attempted to modify the figure accordingly to this suggestion, but the additional text made the figure too cluttered.

6. *Page 4, line 78: Adjust text to “...highly heritable, common, and benign tumors of the uterus with poorly understood etiology.”*

This change has been made.

7. *Page 5, line 106: Remove commas and adjust to “...cumulative incidence of fibroids by age is...”*

This change has been made.

8. *Page 5, line 115: Consider “self-selected” in place of “self-identified”*

This change has been made.

9. *Page 7, line 156: Indicate “Supplemental Figure 1a”*

This change has been made.

10. *Page 11, line 276: Make “Supplemental Table 7b” bold text*

This change has been made.

REVIEWERS' COMMENTS

Reviewer #1 (Remarks to the Author):

Thank you to the authors for thoroughly addressing all comments.

Minor suggestion: Suggest adding the reference for self-reported UL and their finding that 96% was validated in the paper.

Reviewer #2 (Remarks to the Author):

I have no new comments.

Reviewer #3 (Remarks to the Author):

I am satisfied with the revisions the authors have made.

Reviewer #4 (Remarks to the Author):

Reviewer #1:

Minor suggestion: Suggest adding the reference for self-reported UL and their finding that 96% was validated in the paper

Response:

We have added this explanation and reference into our methods statement as the following:
“Some of the phenotyping was based on survey response. Previous literature has shown that self-report can still be a useful approach, with one study validating 96% of self-reported diagnoses”¹

1. Wise, L.A., Palmer, J.R., Stewart, E.A. & Rosenberg, L. Age-specific incidence rates for self-reported uterine leiomyomata in the Black Women's Health Study. *Obstetrics & Gynecology* **105**, 563-568 (2005).